# Validity and Reliability of 2D Video Analysis for Swimming Kick Start Kinematics

**DOI:** 10.3390/jfmk10020184

**Published:** 2025-05-21

**Authors:** Ivan Matúš, Bibana Vadašová, Tomáš Eliáš, Łukasz Rydzik, Tadeusz Ambroży, Wojciech Czarny

**Affiliations:** 1Faculty of Sports, Universtiy of Presov, 08001 Prešov, Slovakia; ivan.matus@unipo.sk (I.M.); bibana.vadasova@unipo.sk (B.V.); tomas.elias@unipo.sk (T.E.); wojciech.czarny@unipo.sk (W.C.); 2Department of Sport Theory and Motor Skills, Institute of Sport Sciences, University of Physical Culture in Kraków, 31-571 Kraków, Poland; tadek@ambrozy.pl; 3Institute of Physical Culture Studies, College of Medical Sciences, University of Rzeszow, 35-310 Rzeszow, Poland

**Keywords:** swimming start biomechanics, 2D kinematic analysis, kick start, reliability, validity, video motion analysis

## Abstract

Background: Objective evaluation of the swimming start is crucial for sprint performance improvement. Traditional visual assessment of its phases—reaction, take-off, flight, and underwater glide—lacks precision. This study addresses the need for more integrated and accessible biomechanical tools by validating IQ LAB software (Version 250319), which is embedded in the SwimPro system and enables immediate video-based motion analysis without external processing. Existing tools like Dartfish (ProSuite 4.0) require separate video handling and licensing, whereas IQ LAB offers a built-in, streamlined solution suitable for applied environments. Methods: We evaluated the concurrent validity of the IQ LAB software, a desktop 2D motion analysis tool, using Dartfish (ProSuite 4.0) as the gold standard. The reliability was assessed using intrarater temporal stability and interrater agreement, considering selected kinematic parameters related to the swimming kick start (to 5 m). A sample of 13 competitive male swimmers (age 17.2 ± 1.1 years) was analyzed across two sessions. Concurrent validity was assessed by comparing IQ LAB outputs to those from Dartfish software in the sagittal plane. Intrarater reliability was measured using a test–retest design across two sessions spaced 7 days apart. Interrater reliability involved two independent raters analyzing the same video data using IQ LAB. Results: IQ LAB and Dartfish kinematic parameters demonstrated strong agreement (Pearson r ≥ 0.95), with no significant systematic differences. The intrarater and interrater reliability were excellent (ICC ≥ 0.94, 95% CI included). The test–retest reliability of the selected parameters across seasons also showed excellent reproducibility (ICC ≥ 0.93). Conclusions: IQ LAB software provides a valid and reliable 2D kinematic assessment of the swimming kick start, offering a practical and accessible tool for coaches and researchers. This study introduces a novel validated software solution for biomechanical analysis in swimming starts.

## 1. Introduction

Starting in swimming can have a significant impact on the overall ranking of swimmers, particularly in sprint events [1]. Given that the time difference for the fastest event of the 50 freestyle at the Tokyo 2020 Olympic Games was 0.72 s (difference in block time—0.08 s) for men and 0.60 s (difference in block time—0.16 s) for women [2], it is crucial to have a comprehensive understanding of all its phases. To gain a more comprehensive understanding of the overall swimming race (start, free swim, and finish), biomechanical analysis can be applied. Within the start phase itself, two sub-phases are typically identified: the above-water and the underwater phase. Several studies have highlighted the performance in these phases and their overall influence on start performance, using different systems and devices to evaluate the kinematic parameters [3,4,5,6]. Currently, biomechanical analysis of movement is utilized in all sports and at different performance levels. Instruments and devices utilizing their software can perform two-dimensional (2D) or three-dimensional (3D) kinematic analysis [7]. Two-dimensional kinematic analysis is considerably less expensive and more accessible. The utilization of 2D kinematic analysis is considerably more straightforward, yet the applicability of the measured data is contingent upon its reliability and validity. Some studies have demonstrated a high correlation between 2D and 3D motion analysis [7,8,9]. Two-dimensional kinematic analysis uses cameras and software to evaluate the kinematic parameters obtained. The cameras can independently upload the video footage to a computer, and then, with the help of the software, the footage can be evaluated. They can also form a system, which includes software that allows the measured parameters to be evaluated instantaneously. Some studies dealing with swimming starts have used a force plate system in conjunction with 2D kinematic analysis [4,5,10,11,12]. As these studies had to adapt the force plate to the OSB11, OSB14, or OSB12 starting blocks, the data obtained from the force plate and 2D kinematic analysis may differ. Nevertheless, force plates are too expensive, and therefore, most studies on swimming starts only evaluate a 2D kinematic analysis of swimmers [13,14,15,16]. In 2D kinematic analyses of starts, anthropometric points are applied to the swimmers’ bodies using waterproof markers, which are then used for its evaluation. When reliability is assessed, data from 2D kinematic analysis are typically evaluated by at least two independent examiners [17]. Some top swimming events (European Championships) have also been evaluated using video analysis, where different software has been used to evaluate the measured parameters [18,19,20]. In the case of more advanced biomechanical analysis, 3D systems or force plates have been used in swimming-specific studies [21,22,23], though these setups are less accessible for routine evaluation. Various programs are used to evaluate the 2D kinematic analysis of the swimming start, such as Dartfish Motion Analysis Software [3,24], Visio version 2021 [25], InThePool (STT Systems, San Sebastián, Spain) [26], and Skill Spector version 1.3 [27]. Despite the frequent use of 2D kinematic analysis of the swimming start in the sagittal plane, few studies have reported its reliability for specific parameters. For instance, Dartfish motion analysis software has been demonstrated in multiple studies [17,28] to be reliable and valid.

While Dartfish has been widely validated, the reliability of the IQ LAB system with the SwimPro camera setup during swimming starts remains underexplored. A major advantage of IQ LAB is its seamless integration into the SwimPro camera system, which allows for real-time kinematic analysis immediately after recording, without the need for external video handling or processing. Unlike Dartfish, which requires external software installation and manual video import, IQ LAB is pre-integrated into the camera workflow, reducing user burden and offering a more cost-effective and practical solution for coaches and researchers. The 5 m distance was selected based on previous research [6,21,22,24], as it represents the end of the underwater glide phase and is critical for analyzing start-specific performance parameters in sprint events. The aim of this study was to assess the reliability and validity of selected kinematic parameters of a 2D video analysis of a 5 m start using IQ LAB software and video footage recorded with the SwimPro system.

## 2. Materials and Methods

The research sample consisted of 13 competitive male swimmers (mean age 17.2 ± 1.1 years, body height 182.6 ± 3.6 cm, and weight 78.3 ± 2.8 kg). The average sprint performance in the 50 m freestyle was 590 ± 38 World Aquatics points. To eliminate potential gender-based differences in hip and knee kinematics, only male participants were included in this study. Individuals who were injured, had problems with their lower limbs, or were post-injury were excluded from this study. Prior to data collection, each participant was informed of the process and purpose of this study and provided written informed consent. This study was conducted in accordance with the ethical standards of the University of Presov, Slovakia, Ethical Committee (Approval No. ECUP042022PO).

Procedure

Testing was conducted in a 25 m pool (6 lanes, depth 1.56–1.86 m) at the University of Prešov. Since the pool lacked a built-in OSB starting block, an official OSB12 platform (345.507.2, Version 1.4, Swiss Timing Ltd.) was mounted on the standard block. The testing was carried out over two sessions (first and second test), with a 7-day interval between them. The start signal was provided by verbal commands (“take your marks”), followed by the activation of a mobile timing device equipped with an LED light. This device was clearly visible on the above-water cameras and served as the timing reference. Following the methodology of Matúš et al., 2021 and 2024 [6,24], waterproof markers were applied bilaterally to anatomical points: lateral margin of the left and right transverse tarsal joint, lateral left and right malleolus, lateral left, and right knee condyle, left and right greater trochanter, lateral margin of the left and right scapular spine, lateral left and right elbow epicondyle, ulnar styloid process of the left and right wrist, and medial side of the 5th metacarpal-phalangeal joint. Although only the right side was analyzed (sagittal plane), bilateral marking was required due to the camera setup (two lateral cameras 6 m apart) to capture both limbs at rest. All markers were placed by the same experienced medical doctor using a standardized anatomical reference protocol. Before the measurement, swimmers performed a warm-up following the RAMP protocol [29], followed by a 400 m swim under the supervision of a swimming coach. After the swim, each swimmer performed two starts according to their preferred position. After completing the start, testing began. Starts were performed in random swimmer order (not randomized per trial). Each swimmer performed two starts per session, and the start with a shorter time of 5 m (T5) was selected for analysis. A 9 min rest was given between attempts, and a 45 s interval between swimmers was ensured to allow completion of a start and adjustment of the kick plate by the next swimmer. According to the study by Tor et al. [4], swimmers were instructed to glide passively up to the 6.6 m mark. No dolphin or flutter kicks were allowed until this distance was reached. Starts that did not meet inclusion criteria—such as premature take-off, arm strokes, or visible leg movement before take-off—were discarded and repeated. The movement on the block before take-off (reaction and push-off phase) lasted approximately 0.5–0.7 s, followed by flight and glide phases.

Instrumentation

We used an OSB12 starting platform (dimensions 740 × 520 × 38 mm) with an adjustable rear support (5 levels over a 200 mm range at a 30° angle). Kinematic parameters were assessed using Dartfish ProSuite 4.0 (Fribourg, Switzerland) and IQ LAB software (version 250319). Video footage was captured using a four-camera SwimPro system, recording at 50 Hz and shutter speed of 1/1000 s. Two lateral cameras were spaced 6 m apart, with the OSB12 platform between them, to ensure visibility of both limbs at rest. One camera was positioned 1.6 m from the pool edge at a height of 1.5 m. One underwater camera was placed 5 m from the pool wall at a depth of −1.7 m. LED lighting was used to enhance visibility. Calibration was performed using a fixed reference frame visible in all camera views. A schema of the camera layout is provided in Figure 1.

Concurrent Validity

Based on the anatomical points, the following parameters were analyzed in the sagittal plane using both Dartfish and IQ LAB software: Angles: front and rear ankle angle (FAA, RAA), front and rear knee angle (FKA, RKA), hip angle (HA), take-off angle (TA), and entry angle (EA); times: block time (BT), flight time (FT), glide time (GT), and time to 5 m (T5); distances: flight distance (FD) and glide distance (GD) Although front and rear joint angles are geometrically complementary, both were measured to detect possible asymmetries. All video digitization was performed manually by trained evaluators. To minimize variability, marker placement followed a standardized anatomical reference protocol. A second trial (with shorter T5) from the second test session was used to assess concurrent validity. Pearson’s correlation, paired-sample *t*-tests, and Bland–Altman plots with confidence intervals were used to compare outputs.

Intrarater and Interrater Reliability

Intrarater reliability was assessed by Evaluator A, who reanalyzed all 13 trials after a 7-day interval. Interrater reliability was assessed between Evaluators A and B on the same 13 trials. Intraclass correlation coefficients (ICC) and standard error of measurement (SEM) were calculated using single-rating, absolute-agreement, two-way random-effects models.

Test–Retest Reliability

Test–retest reliability was determined by comparing the best trial (shortest T5) from the first and second testing sessions. All analyses were performed by Evaluator A using a consistent protocol. ICC and SEM were calculated for all selected variables.

Statistical Analysis

In the 2D kinematic analysis of starts, we processed the mean (M), standard deviation (SD), and standard error of mean (SEM) of the selected kinematic parameters. Concurrent validity was assessed using Pearson’s correlation and two-tailed paired *t*-tests. Bland–Altman plots were used to compare Dartfish and IQ LAB outputs. Intraclass correlation coefficients (ICC) were used to assess intrarater and interrater reliability between evaluators and between testing sessions. ICC values were interpreted based on Fleiss [30]: low (<0.40), moderate (0.40–0.75), substantial (0.75–0.90), and excellent (>0.90). SEM was calculated from ICC values [31]. Statistical analysis was conducted using IBM SPSS (v27.0.1.0). Microsoft Excel (2019) was used to record and organize kinematic parameters exported from Dartfish and IQ LAB software.

## 3. Results

This study found excellent agreement between the Dartfish and IQ LAB software across all evaluated kinematic parameters of the swimming kick start. The results showed high correlations (r = 0.98–0.99), no systematic bias in Bland–Altman plots, and excellent intrarater, interrater, and test–retest reliability (ICC = 0.93–1.00) with low SEM values (0.00–0.40). Table 1 provides the mean values (M), standard error of the mean (SEM), and standard deviations (SD) for selected kinematic parameters during the kick start from OSB12 at a 5 m distance, recorded using SwimPro video cameras. These parameters were evaluated using Dartfish and IQ LAB software. Correlations between the two software outputs ranged from 0.98 to 0.99, indicating excellent agreement. There were no statistically significant differences between the evaluated kinematic parameters as determined by paired-sample *t*-tests (*p* > 0.05) (Table 1). Mean differences and 95% confidence intervals (CI) between software outputs are presented in Table 1 and illustrated in Bland–Altman plots (Appendix A). In the Bland–Altman plots, one outlier was observed for the front knee angle (FKA) and flight time (FT), and two outliers were found for the flight distance (FD). However, the plots revealed no systematic bias, supporting the consistency between the Dartfish and IQ LAB analyses.

Table 2 and Table 3 present descriptive statistics and intrarater and interrater reliability values for the 2D kinematic analysis of repeated measurements. Intrarater and interrater ICC values for the evaluated parameters were excellent (ICC ≥ 0.97), with SEM ranging from 0.00 to 0.30 (Table 3). Table 4 displays descriptive values across test sessions 1 and 2. **The test**–retest reliability for the selected kinematic parameters was excellent, with ICC values ranging from 0.93 to 0.99 and SEM from 0.01 to 0.40.

## 4. Discussion

The aim of this study was to assess the reliability and validity of selected kinematic parameters of a 2D video analysis of the swimming kick start to 5 m using IQ LAB software in the sagittal plane. Currently, various instruments and systems are available for identifying key performance-related parameters in the swimming start [5,24,32,33,34]. However, the selection of suitable equipment is often constrained by financial resources. The ideal system should provide immediate feedback while ensuring objectivity, validity, and reliability. Previous studies, such as those by Tor et al. [4], Silveira et al. [5], Matúš et al. [24], and Burkhardt et al., 2023 [35], have identified important kinematic parameters in swim starts. Yet, few have reported the reliability of the derived values obtained from such systems. In our study, concurrent validity was tested by comparing IQ LAB and Dartfish outputs for selected start parameters. High correlation coefficients (r ≥ 0.95) and non-significant differences confirmed strong concurrent validity. Additionally, excellent intrarater and interrater reliability indicated that IQ LAB provides consistent angle and time measurements across evaluators within a single session. The SwimPro system, combined with IQ LAB, offers the advantage of capturing high-quality video suitable for motion analysis from block to 5 m. However, one limitation is that the digitization of joint points must be performed manually in each video frame, unlike in Dartfish where some automation is available. Manual digitization, though time-consuming, may allow for more precise point placement. The excellent ICC values may be attributed to consistent camera positioning and controlled testing conditions, though manual digitization could still introduce minor human error. Consistent setup and standardized testing protocols likely contributed to the high reliability despite these limitations. Studies using Dartfish software have shown strong concurrent validity and interrater reliability in sagittal plane joint angle analysis [17,28]. Systems such as Wetplate or Dartfish depend on visual clarity and lighting to ensure accuracy; in suboptimal conditions, manual corrections are often required [4]. Similarly, manual methods were used in our study, and all raters were trained to ensure consistency. Reliability estimates in our data were excellent across all parameters, with test–retest ICC values ranging from 0.90 to 0.99. Slightly lower reliability between testing sessions may be attributed to subtle differences in swimmer technique or small inconsistencies in marker placement between days. Despite this, strict protocol adherence and controlled recording conditions helped minimize error and sustain measurement reliability. These results reinforce that the SwimPro + IQ LAB setup is a reliable tool for kinematic analysis in swimming.

Limitations

This study has several limitations. First, the sample size was small (*n* = 13), and all participants were male competitive swimmers. Although homogeneous in skill and health status, this limits the generalizability of the results. Future studies should include larger and more diverse cohorts, including female participants and a broader age range. Second, manual digitization of anatomical points introduces potential for human error, even though standardized protocols and visible markers were used. While marker placement was performed consistently by a trained medical professional, minor variations may have occurred and contributed to slight measurement differences. Despite these limitations, the findings demonstrate that IQ LAB software with SwimPro cameras delivers reliable and valid measurements of swim start kinematics in controlled conditions. This makes the system promising for both research applications and practical use in coaching environments.

## 5. Conclusions

This study demonstrated that the combination of SwimPro video capture and IQ LAB software offers a valid and reliable approach to the 2D kinematic analysis of swimming starts in the sagittal plane. High agreement with Dartfish software confirmed strong concurrent validity, while intra- and interrater reliability values indicated robust measurement consistency across evaluators. The system’s practicality and low cost make it a promising tool for applied sports environments. These findings support the use of IQ LAB software for accessible biomechanical feedback in performance settings, although future validation in larger, more diverse cohorts is recommended. The continued development and refinement of such systems may enhance the integration of video-based analysis into routine coaching and athlete development.

## Figures and Tables

**Figure 1 jfmk-10-00184-f001:**
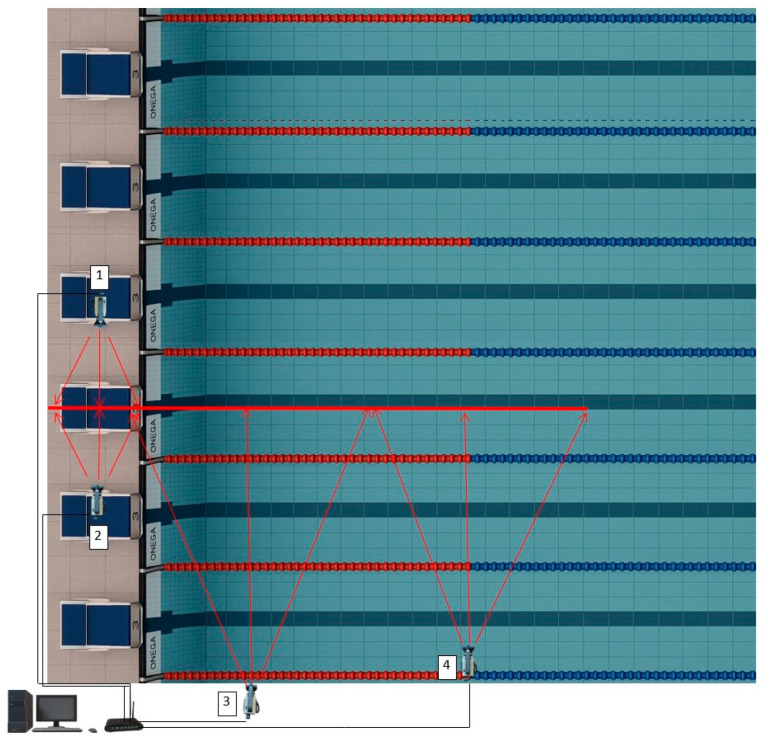
Set-up of the cameras. The red line represents the swimmer’s movement relative to the cameras. The red lines from the camera are the fields of view of the cameras.

**Table 1 jfmk-10-00184-t001:** Dartfish and IQ Lab measurements of kinematic parameters in kick start to 5 m.

Variables	Dartfish			IQ LAB			Pearson Coefficient	Paired-Sample T-Test							
											95% CI				Sig. (2-Tailed)
	M	SEM	SD	Mean	SEM	SD		M	SEM	SD	Lower	Upper	t	df	
FKA (°)	132.24	0.62	2.22	132.23	0.61	2.21	0.998 **	0.01	0.03	0.13	−0.07	0.08	0.22	12.00	0.83
FAA (°)	126.91	0.96	3.48	126.92	0.95	3.41	0.999 **	−0.02	0.04	0.13	−0.09	0.06	−0.43	12.00	0.67
RKA (°)	84.16	0.45	1.64	84.12	0.45	1.61	0.998 **	0.05	0.03	0.11	−0.02	0.11	1.48	12.00	0.17
RAA (°)	100.08	0.62	2.24	100.10	0.63	2.28	0.999 **	−0.02	0.03	0.12	−0.09	0.05	−0.71	12.00	0.49
HA (°)	44.97	0.32	1.15	44.98	0.32	1.16	0.997 **	−0.01	0.02	0.09	−0.06	0.04	−0.32	12.00	0.75
BT (s)	0.79	0.01	0.01	0.79	0.01	0.01	0.997 **	0.00	0.00	0.00	0.00	0.00	1.90	12.00	0.08
TA (°)	39.53	0.28	1.00	39.52	0.28	1.1	0.997 **	0.01	0.02	0.08	−0.04	0.05	0.37	12.00	0.72
EA (°)	36.62	0.21	0.75	36.65	0.21	0.74	0.995 **	−0.02	0.02	0.07	−0.07	0.02	−1.15	12.00	0.27
FT (s)	0.40	0.01	0.01	0.40	0.01	0.01	0.997 **	0.0	0.00	0.00	0.00	0.00	−0.81	12.00	0.44
FD (m)	2.82	0.02	0.05	2.82	0.01	0.05	0.997 **	0.00	0.00	0.00	0.00	0.00	−1.48	12.00	0.17
GT (s)	0.56	0.01	0.01	0.56	0.01	0.01	0.995 **	0.00	0.00	0.00	0.00	0.00	0.27	12.00	0.79
GD (m)	2.18	0.01	0.05	2.18	0.01	0.05	0.997 **	0.00	0.00	0.00	0.00	0.00	1.48	12.00	0.17
T5 (s)	1.75	0.01	0.01	1.75	0.01	0.01	0.997 **	0.00	0.00	0.00	0.00	0.00	0.46	12.00	0.66

Note—M—means, SEM—standard error of the means, SD—standard deviations, FKA (°)—front knee angle, FAA (°)—font ankle angle, RKA (°)—rear knee angle, HA (°)—hip angle, BT (s)—block time, TA (°)—take-off angle, entry angle (EA), FT (s)—flight time, FD (m)—flight distance, GT (m)—glide time, GD (m)—glide distance, and T5 (s)—time to 5 m. ** *p* < 0.01.

**Table 2 jfmk-10-00184-t002:** Two-dimensional kinematic analysis by IQ Lab for intrarater and interrater measurements from kick start to 5 m.

Variables	Intrarater						Interrater					
	Ranking 1			Ranking 2			Examiner A			Examiner B		
	M	SEM	SD	M	SEM	SD	M	SEM	SD	Mean	SEM	SD
FKA (°)	132.58	0.54	1.94	132.48	0.54	1.95	132.58	0.54	1.94	132.54	0.50	1.82
FAA (°)	126.96	0.86	3.12	126.88	0.87	3.12	126.96	0.86	3.12	127.17	0.82	2.96
RKA (°)	84.35	0.44	1.60	84.32	0.44	1.60	84.35	0.44	1.60	84.43	0.46	1.66
RAA (°)	100.58	0.66	2.38	100.60	0.67	2.41	100.58	0.66	2.38	100.70	0.70	2.51
HA (°)	44.97	0.32	1.15	44.98	0.32	1.16	44.97	0.32	1.15	45.12	0.36	1.30
BT (s)	0.79	0.01	0.01	0.79	0.01	0.01	0.79	0.01	0.01	0.79	0.01	0.01
TA (°)	39.63	0.28	1.02	39.62	0.28	1.02	39.63	0.28	1.02	39.54	0.26	0.93
EA (°)	36.92	0.19	0.67	36.92	0.19	0.67	36.92	0.19	0.67	36.87	0.19	0.70
FT (s)	0.40	0.01	0.01	0.40	0.01	0.01	0.40	0.01	0.01	0.40	0.01	0.01
FD (m)	2.84	0.01	0.05	2.83	0.01	0.05	2.84	0.01	0.05	2.84	0.01	0.05
GT (s)	0.56	0.01	0.01	0.56	0.01	0.01	0.56	0.01	0.01	0.56	0.01	0.01
GD (m)	2.16	0.01	0.05	2.17	0.01	0.05	2.16	0.01	0.05	2.16	0.01	0.05
T5 (s)	1.75	0.01	0.01	1.75	0.01	0.01	1.75	0.01	0.01	1.74	0.01	0.02

Note—M—means, SEM—standard error of the means, SD—standard deviations, FKA (°)—front knee angle, FAA (°)—font ankle angle, RKA (°)—rear knee angle, HA (°)—hip angle, BT (s)—block time, TA (°)—take-off angle, entry angle (EA), FT (s)—flight time, FD (m)—flight distance, GT (m)—glide time, GD (m)—glide distance, and T5 (s)—time to 5 m.

**Table 3 jfmk-10-00184-t003:** Intrarater and interrater reliability measurements between rankings and raters by IQ Lab 2D kinematic analysis from kick start to 5 m.

Variables	Intrarater Reliability					Interrater Reliability			
	ICC				Raters	ICC			
	IC	95% CI		SEM		IC	95% IC		SEM
		Lower	Upper				Lower	Upper	
FKA (°)	0.99	0.98	1.00	0.19	A-B	0.99	0.98	1.00	0.19
FAA (°)	1.00	0.99	1.00	0.00	A-B	0.99	0.97	1.00	0.30
RKA (°)	1.00	1.00	1.00	0.00	A-B	0.99	0.95	1.00	0.16
RAA (°)	1.00	1.00	1.00	0.00	A-B	1.00	0.99	1.00	0.00
HA (°)	1.00	1.00	1.00	0.00	A-B	0.97	0.90	0.99	0.21
BT (s)	1.00	0.99	1.00	0.00	A-B	0.98	0.93	0.99	0.01
TA (°)	1.00	1.00	1.00	0.00	A-B	0.99	0.97	1.00	0.01
EA (°)	0.99	0.97	1.00	0.01	A-B	0.97	0.90	0.99	0.12
FT (s)	1.00	1.00	1.00	0.00	A-B	0.97	0.89	0.99	0.01
FD (m)	0.99	0.98	1.00	0.01	A-B	0.99	0.96	1.00	0.01
GT (s)	1.00	1.00	1.00	0.00	A-B	0.94	0.80	0.98	0.01
GD (m)	0.99	0.98	1.00	0.01	A-B	0.99	0.96	1.00	0.01
T5 (s)	1.00	1.00	1.00	0.00	A-B	0.97	0.91	0.99	0.01

Note— SEM—standard error of the means, SD—standard deviations, ICC—intraclass correlation coefficients, IC—intraclass correlation coefficient, CI—confidence interval, FKA (°)—front knee angle, FAA (°)—font ankle angle, RKA (°)—rear knee angle, HA (°)—hip angle, BT (s)—block time, TA (°)—take-off angle, entry angle (EA), FT (s)—flight time, FD (m)—flight distance, GT (m)—glide time, GD (m)—glide distance, and T5 (s)—time to 5 m.

**Table 4 jfmk-10-00184-t004:** Reliability of kinematic parameters measured between two test sessions by IQ Lab 2D kinematic analysis from kick start to 5 m.

Variables	Test Session 1			Test Session 2			ICC			
	M	SEM	SD	M	SEM	SD	IC	95% CI		SEM
								Lower	Upper	
FKA (°)	132.40	0.62	2.23	132.53	0.54	1.93	0.98	0.93	0.94	0.29
FAA (°)	126.98	0.92	3.33	126.92	0.86	3.11	0.99	0.96	0.99	0.32
RKA (°)	84.20	0.44	1.60	84.34	0.44	1.60	0.99	0.97	0.99	0.16
RAA (°)	100.09	0.63	2.26	100.59	0.66	2.39	0.97	0.89	0.99	0.40
HA (°)	44.94	0.33	1.18	44.83	0.32	1.17	0.98	0.94	0.99	0.17
BT (s)	0.79	0.01	0.01	0.79	0.01	0.01	0.94	0.79	0.98	0.01
TA (°)	39.55	0.28	1.00	39.63	0.28	1.02	0.97	0.90	0.99	0.25
EA (°)	36.69	0.19	0.67	36.92	0.19	0.67	0.95	0.85	0.99	0.15
FT (s)	0.40	0.01	0.01	0.40	0.01	0.01	0.90	0.67	0.97	0.01
FD (m)	2.82	0.01	0.05	2.83	0.01	0.05	0.94	0.81	0.98	0.01
GT (s)	0.55	0.01	0.02	0.56	0.01	0.01	0.93	0.76	0.98	0.01
GD (m)	2.18	0.01	0.05	2.17	0.01	0.05	0.94	0.81	0.98	0.01
T5 (s)	1.74	0.01	0.02	1.75	0.01	0.01	0.96	0.87	0.99	0.01

Note—M—means, SEM—standard error of the means, SD—standard deviations, ICC—intraclass correlation coefficients, IC—intraclass correlation coefficient, CI—confidence interval, FKA (°)—front knee angle, FAA (°)—font ankle angle, RKA (°)—rear knee angle, HA (°)—hip angle, BT (s)—block time, TA (°)—take-off angle, entry angle (EA), FT (s)—flight time, FD (m)—flight distance, GT (m)—glide time, GD (m)—glide distance, and T5 (s)—time to 5 m.

## Data Availability

The raw data supporting the conclusions of this article will be made available by the authors, without undue reservation.

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
