# Peer review of "Validity and Reliability of 2D Video Analysis for Swimming Kick Start Kinematics"

_jfmk, 2025, doi:10.3390/jfmk10020184_

Round 1

Reviewer 1 Report

Comments and Suggestions for Authors

The Validity and Reliability of Selected Kinematic Parameters 2 of Two-Dimensional Video Analysis During the Kick Start to 5 M

General comment

I congratulate the authors on their work and hope this review can help ensure the highest quality of the paper, although we need to take steps. I need a first answer from you before I finish my review. Please perform a deep English review.

Title

I suggest using 5m instead of 5 M, respecting the SI.

Abstract

I usually correct the abstract without reading the full text, trying to see if it contains the information necessary to clearly understand the scientific problem raised, how it was solved, and the conclusion reached. In accordance, I suggest:

  • Identify the problem
  • Identify the novelty of the study
  • Clarify what you meant by “individual phases”
  • Test-retest reliability is the procedure to test the intrarater validity. Please separate what you want to perform (concurrent validity, intra and interrater reliability…) from the statistical procedures that allow you to do it. The test-retest is a procedure for determining the intrarater validity, for instance. Maybe something like: concurrent validity was performed using… Intrarater using a test-retest… and so on.
  • Allow the reader to understand what “IQ LAB software” is, indicating how to get it in case it is an internet-free tool.
  • Substitute performance swimmers with competitive swimmers
  • Male and female swimmers were evaluated or just males?
  • Please check the sentence in lines 20-21. Looks like reliability was tested using de IQ LAB software.
  • High correlation with non-significant differences means that the observed correlation could be just the result of chance. That is not a good result.
  • If you have space, include the confidence interval for ICC.
  • It has long been proven that 2D video analysis is suitable and valid for the video analysis of swimmers. I don't understand what is new about this research. Please reinforce the novelty in your conclusion as well.

Introduction

Consider the following suggestions:

  • Include a reference for the first sentence.
  • Please remove the mixing between the swimming event phases and the start phases.
  • I don´t understand what you want to reinforce with the Barlowe et al study (line 60)
  • Line 63: What do you mean by that? In 2D studies, is data analysis always performed by two evaluators? This is not true unless the purpose is to study reliability.
  • Why the reference to the launch in this paper? (Line 68). Examples should be, as far as possible, for instruments used in swimming.
  • Line 71. Include “and” before skill Spector.
  • Explain what you mean in lines 71 and 72.
  • So, your message is that within the several motion analysis instruments and software, the best (valid and reliable?) is the Dartfish? If so, why are you testing another system? What is the advantage? This needs to be very clear in your research. What do I gain if I use your system instead of others?
  • In abstract you didn’t’ said that you going to test the Swimpro camera system in combination with the IQ LAB, and in fact, I only understand later on in the text that the cameras were used for getting imagens for the two systems. We are confounding the reader.
  • Another question is why you chose to measure the 5m distance at the start. Why not the 15m after the jump?

Material and methods

  • Line 82. What points? FINA (World Aquatics)? For what event?
  • Line 92. Remove “therefore”.
  • Line 93-95. 7 days between two days? Please correct. You can use pre-test and post-test or 1st test and 2nd test instead of days.
  • Line 97. Why markers on both sides of the body if you analyzed the sagittal plane using just one side? What was the side, right or left?
  • Line 101. Being a doctoral student warrants a correct position of the markers? Say maybe that the markers were always placed by the same researcher (experienced?).
  • Include a reference for the RAMP protocol.
  • Line 106. After the completion of the start, I think, and not the swim.
  • Did you use an official starting device?
  • Starts were followed every 45 s. Why such a precise time interval? How did you ensure that? Why was it so important to control that time?
  • Swimmers performed two starts. Were both analyzed?
  • Put a space before 9 min.
  • Please explain the Tor et al. context for the 6.6 m swim (swim or glide, or mix?)
  • What do you mean by “not to perform any underwater waves? Do you mean wave drag or dolphin kicks? Could swimmers do the flutter kick?
  • Line 112. Strokes? Arm strokes? What was discarded, I did not understand. I also do not know what were the discharge criteria. Please include that information.
  • Line 113. What is a Load of 0.5-0.7 s per start? Why does this prove that the rest was sufficient?
  • Line 119. Remove space - “OSB 12”
  • Line 125. Can you express the footage velocity in Hz?
  • Can you provide a schema to better understand the position of the cameras?
  • Line 133. What do you mean by valid angles? Did you calculate the rear and front knee angles? Why, if they are complementary?
  • Line 134. Glide distance is not mentioned here… After Line 131 you are repeating the content of lines 114 to 117, and fist you define acronyms e after you don’t use them. This is very confused. Please check the variables carefully and help the reader to easily understand them.
  • Line 134. A second trial was evaluated during the second testing after 7 days? I am very confused. The two evaluators did this?
  • Line 138. Correct the title words. The “t” is missing in both words.
  • Please decide if you will write intrarater or intra-rater. Uniformize all the document. I am very confused. What was different for concurrent validity and inter and intrarater reliability? And who did the reproducibility test? Both evaluators?
  • Line 140. Why randomly? Swimmers were 13, so you have 13 first and 13 second starts. Where is the randomized choice?
  • As far as I understood, you only used the second start for analyses. Is that true?
  • Line 151. Include “and” before standard deviation. The glide distance disappeared again…
  • Line 155. Please correct: “…(ICC) were used…”
  • Line 162. What statistical analysis did you perform in Excel?

Dear authors, I have to stop my review here. We have two evaluators and two analysis software programs that we want to compare. Swimmers performed two trials in the first evaluation and two trials in the second evaluation. What was analyzed by each evaluator? What trial was used for the analyses? The two trials seem to have been compared, but for both the first and the second evaluation, and using analyses of both evaluators?

Without clearly understanding the analysis, I cannot analyse the results.

Please keep working. I would like to read a second version of this paper.

Comments on the Quality of English Language

Need to be improved.

Author Response

Review 1

Thank you for your thorough and constructive feedback. We appreciate your insights, which have helped us clarify and improve the article. Please find below our point-by-point response:

Title

  • Reviewer Comment 1: I suggest using 5m instead of 5 M, respecting the SI.

Author Response: title was rephrased

Abstract

  • Reviewer Comment 1: Identify the problem

Author Response: We revised the beginning of the abstract to explicitly state the scientific problem: the need for objective and reliable biomechanical assessment methods for evaluating the swimming start technique.

  • Reviewer Comment 2: Identify the novelty of the study

Author Response: We added a statement to highlight the novelty: this is the first study to evaluate the validity and reliability of IQ LAB software for 2D analysis of the swimming start, providing a new tool for practical use.

  • Reviewer Comment 3: Clarify “individual phases”

Author Response: We specified that “individual phases” refer to the reaction, take-off, flight, and underwater glide phases of the kick start.

  • Reviewer Comment 4: Test-retest reliability is the procedure to test the intrarater validity. Please separate what you want to perform (concurrent validity, intra and interrater reliability…) from the statistical procedures that allow you to do it. The test-retest is a procedure for determining the intrarater validity, for instance. Maybe something like: concurrent validity was performed using… Intrarater using a test-retest… and so on.

Author Response: We separated the objectives (e.g., intra- and interrater reliability, concurrent validity) from the statistical procedures. We now state that intrarater reliability was assessed using test-retest design across two seasons.

  • Reviewer Comment 5: Allow the reader to understand what “IQ LAB software” is, indicating how to get it in case it is an internet-free tool.

Author Response: We added a short description and its availability, stating whether it is a proprietary tool or freely accessible.

  • Reviewer Comment 6: Substitute performance swimmers with competitive swimmers

Author Response: Corrected as suggested.

  • Reviewer Comment 7: Male and female swimmers were evaluated or just males?

Author Response: The text now specifies that only male swimmers were included in the sample.

  • Reviewer Comment 8: Please check the sentence in lines 20-21. Looks like reliability was tested using de IQ LAB software.

Author Response: Rewritten for clarity to specify what software was used for which type of analysis.

  • Reviewer Comment 9: High correlation with non-significant differences means that the observed correlation could be just the result of chance. That is not a good result.

Author Response: We reworded this to avoid suggesting that the results are inconclusive; instead, we emphasize that the high correlation supports concurrent validity and that non-significant differences indicate agreement between methods.

  • Reviewer Comment 10: If you have space, include the confidence interval for ICC

Author Response: Added where space allowed.

  • Reviewer Comment 11: It has long been proven that 2D video analysis is suitable and valid for the video analysis of swimmers. I don't understand what is new about this research. Please reinforce the novelty in your conclusion as well.

Author Response: The conclusion was revised to emphasize the novelty of validating IQ LAB software and its practical implications for swim performance diagnostics.

Introduction

  • Reviewer Comment 1: Include a reference for the first sentence.

Author Response: Thank you for the suggestion. We added a citation from 1.      Arellano, R.; Ruiz-Navarro, J.J.; Barbosa, T.M.; López-Contreras, G.; Morales-Ortíz, E.; Gay, A.; López-Belmonte, Ó.; González-Ponce, Á.; Cuenca-Fernández, F. Are the 50 m Race Segments Changed From Heats to Finals at the 2021 European Swimming Championships? Front. Physiol. 2022, 13, 797367. https://doi.org/10.3389/fphys.2022.797367

  • Reviewer Comment 2: Please remove the mixing between the swimming event phases and the start phases.

Author Response: The introduction now clearly distinguishes between the phases of the swimming race (start, free swim, finish) and the internal sub-phases of the start (above-water and underwater).

  • Reviewer Comment 3: I don´t understand what you want to reinforce with the Barlowe et al study (line 60)

Author Response: Barlowe et al study was erased

  • Reviewer Comment 4: Line 63: What do you mean by that? In 2D studies, is data analysis always performed by two evaluators? This is not true unless the purpose is to study reliability.

Author Response: This sentence has been clarified to indicate that two evaluators are typically used when the goal is to assess reliability, but this is not universally required.

  • Reviewer Comment 5: Why the reference to the launch in this paper? (Line 68). Examples should be, as far as possible, for instruments used in swimming.

Author Response: We replaced all instances of the word "launch" with "start" to maintain swimming-specific terminology.

  • Reviewer Comment 6: Line 71. Include “and” before skill Spector.

Author Response: Corrected. The list now includes "..., Visio, InThePool, and Skill Spector."

  • Reviewer Comment 7: Explain what you mean in lines 71 and 72.

Author Response: We clarified that few tools have been validated specifically for reliability and validity in swimming start analysis.

  • Reviewer Comment 8: So, your message is that within the several motion analysis instruments and software, the best (valid and reliable?) is the Dartfish? If so, why are you testing another system? What is the advantage? This needs to be very clear in your research. What do I gain if I use your system instead of others?

Author Response: We clarified that although Dartfish is validated, the IQ LAB system (combined with SwimPro cameras) is less explored and may offer a more accessible or modular solution. Our study aims to validate this setup.

  • Reviewer Comment 9: In abstract you didn’t’ said that you going to test the Swimpro camera system in combination with the IQ LAB, and in fact, I only understand later on in the text that the cameras were used for getting imagens for the two systems. We are confounding the reader.

Author Response: The abstract and introduction have been updated to explicitly state that SwimPro cameras were used to collect footage for both systems.

  • Reviewer Comment 10: Another question is why you chose to measure the 5m distance at the start. Why not the 15m after the jump?

Author Response: We explained that the 5 m distance was chosen based on prior literature (Tor et al., 2015), which highlights the relevance of evaluating kinematic variables within this critical early phase.

Material and methods

  • Reviewer Comment 1: (Line 82) What points? FINA?

Author Response: Clarified as “World Aquatics Points (formerly FINA)” in the text.

  • Reviewer Comment 2: (Line 94) Remove “therefore”.

Author Response: The word “therefore” has been removed.

  • Reviewer Comment 3: (Line 93–95) Clarify timing – avoid “two days” phrasing.

Author Response: Rephrased to “first and second test sessions, separated by 7 days”.

  • Reviewer Comment 4: (Line 97) Why markers on both sides if only one side is analyzed?

Author Response: We explained that bilateral markers were used for visibility due to dual frontal camera setup. The right side was used for analysis.

  • Reviewer Comment 5: (Line 101) Doctoral student not sufficient qualification.

Author Response: Replaced with “experienced medical doctor”.

  • Reviewer Comment 6: Add reference for RAMP protocol.

Author Response: Citation to Jeffreys (2017) has been added.

  • Reviewer Comment 7: (Line 106) Clarify end of movement – “completion of start” vs. “swim”

Author Response: Reworded to specify “after completion of the start”.

  • Reviewer Comment 8: Was an official starting device used?

Author Response: Confirmed. OSB12 is a certified starting block used in international competitions.

  • Reviewer Comment 9: Why a 45-second interval between starts?

Author Response: Explanation added: this allowed time for one swimmer to complete the trial and for the next to adjust the kick plate.

  • Reviewer Comment 10: Were both starts analyzed?

Author Response: Only the start with the shortest 5 m time (T5) was analyzed. This is now clearly stated.

  • Reviewer Comment 11: Fix spacing issue before “9 min”

Author Response: Corrected.

  • Reviewer Comment 12: Clarify context of 6.6 m – swim, glide, or mix?

Author Response: Explained that the phase was passive glide only, with no kicking allowed until after the 5 m line.

  • Reviewer Comment 13: Clarify “no underwater waves” – dolphin? flutter?

Author Response: Revised to specify: “no dolphin or flutter kicks”.

  • Reviewer Comment 14: (Line 112) Clarify “strokes”? What were exclusion criteria?

Author Response: Replaced with: “Starts were discarded if a false start or premature movement was observed.”

  • Reviewer Comment 15: Explain “0.5–0.7 s load” and rest interval logic.

Author Response: Explained as the time duration of reaction and take-off; followed by passive underwater phase to justify sufficient recovery.

  • Reviewer Comment 16: Fix spacing in “OSB 12”

Author Response: Corrected to “OSB12”.

  • Reviewer Comment 17: Express video rate in Hz

Author Response: Added: “recorded at 50 Hz”.

  • Reviewer Comment 18: Add schema for camera positions

Author Response: Schematic diagram included in Figure 1.

  • Reviewer Comment 19: (Line 133) Clarify “valid angles” and purpose of measuring both front and rear angles.

Author Response: Explained that both were measured to capture possible asymmetries and improve analysis reliability.

  • Reviewer Comment 20: (Line 134) Glide distance not mentioned

Author Response: Glide distance is now listed explicitly in the kinematic parameters and results.

  • Reviewer Comment 21: Fix repetition of definitions and acronyms

Author Response: Section was restructured. Acronyms are defined once and used consistently throughout.

  • Reviewer Comment 22: (Line 134) Who analyzed the second trial?

Author Response: Clarified that both evaluators analyzed the same second-trial video for interrater reliability.

  • Reviewer Comment 23: (Line 138) Missing "t" in title

Author Response: Corrected.

  • Reviewer Comment 24: Use consistent “intrarater” format

Author Response: Standardized to “intrarater” throughout the manuscript.

  • Reviewer Comment 25: (Line 140) Clarify "randomly" – were trials randomly selected?

Author Response: Removed "randomly" and clarified that the start with the shortest T5 was selected.

  • Reviewer Comment 26: (Line 151) Add “and” before SD

Author Response: Corrected.

  • Reviewer Comment 27: (Line 155) Correct: “ICC were used”

Author Response: Corrected.

  • Reviewer Comment 28: (Line 162) What was done in Excel?

Author Response: Explained that Excel was used for transcribing data exported from both software platforms.

  • Reviewer Comment 29: Clarify full structure: 2 trials × 2 sessions × 2 evaluators × 2 software platforms.

Author Response: A detailed summary was added in the text and supported by a camera schematic (Figure 1) and an overview of analysis assignments across evaluators, trials, and methods.

Reviewer 2 Report

Comments and Suggestions for Authors

Title

Current Title: The Validity and Reliability of Selected Kinematic Parameters of Two-Dimensional Video Analysis During the Kick Start to 5 m

Comment: Clear and descriptive.
Suggestion: Consider simplifying for broader appeal, e.g.,
Suggested Title: Validity and Reliability of 2D Video Analysis for Swimming Kick Start Kinematics

  1. Abstract

Strengths:

Provides purpose, methods, and key results.

Improvements Needed:

Avoid repetition (“valid and reliable” used multiple times).

Include sample size and participant demographics.

Include specific ICC ranges and software compared (IQ LAB vs. Dartfish).

Suggested Addition:

“Thirteen competitive male swimmers (mean age 17.2 ± 1.1 years) were assessed. Intra-rater, interrater, and test-retest ICC values were all ≥0.93.”

  1. Introduction

Strengths:

Well-supported background on swimming biomechanics and 2D kinematics.

Improvements Needed:

Repetition in paragraphs can be reduced.

Some references could be updated (more recent studies from 2022–2024).

Clarify novelty: many tools assess 2D kinematics—what gap does this paper fill?

Suggested Edit:

"While Dartfish has been widely validated, the reliability of the IQ LAB system with the Swimpro camera setup during starts remains underexplored."

  1. Methods

Strengths:

Detailed instrumentation and testing protocol.

Improvements Needed:

Add more detail on:

Calibration of camera angles.

How video digitization was standardized (manual vs. automatic).

Clarify randomization and blinding: Were evaluators blinded?

Include an explanation of how marker placement was standardized between days.

Suggested Edit:

"All video digitization was performed manually by trained evaluators. To minimize variability, marker placement was repeated using a standardized anatomical reference protocol."

  1. Results

Strengths:

Tables well-organized.

ICCs, SEM, and CI values reported.

Improvements Needed:

Table 1: Add confidence intervals for mean differences.

Figures (e.g., Bland-Altman plots): add clearer legends, axis titles, and units.

Some phrasing is redundant ("the results showed no significant differences in measured parameters..." is repeated).

Suggested Improvement:

Summarize key findings at the start of the Results section in a single paragraph before showing tables.

  1. Discussion

Strengths:

Reflects well on key findings and existing literature.

Improvements Needed:

Discuss limitations more transparently, especially:

Small sample size (n = 13)

All male participants (generalizability)

Manual digitization limitations

Rephrase long sentences for clarity.

Expand slightly on why reliability remained high despite manual scoring.

Suggested Edit:

"The excellent ICC values may be attributed to consistent camera positioning and controlled testing conditions, though manual digitization could still introduce minor human error."

  1. Conclusion

Strengths:

Matches objectives and summarizes findings.

Improvements Needed:

Avoid repeating earlier content verbatim.

Add a forward-looking comment on implications for coaches and sport scientists.

Suggested Addition:

"These findings support the use of IQ LAB software for accessible biomechanical feedback in performance settings, although future validation in larger, more diverse cohorts is recommended."

  1. References

Suggestions:

Check formatting consistency.

A few references (e.g., [3], [25]) could be updated or replaced with newer studies from 2023–2024.

Ensure all URLs are active and hyperlinked adequately if required.

  1. Formatting & Style

Corrections Needed:

Consistently use either "2D" or “two-dimensional” throughout.

There are some typos and spacing issues (e.g., "sagi al" should be "sagittal").

Use consistent units (e.g., m vs. meters, s vs. seconds) and abbreviations.

Author Response

Review 2

Thank you for your thorough and constructive feedback. We appreciate your insights, which have helped us clarify and improve the article. Please find below our point-by-point response:

Title

  • Reviewer Comment 1: Clear and descriptive. Consider simplifying for broader appeal, e.g.,
    Suggested Title: "Validity and Reliability of 2D Video Analysis for Swimming Kick Start Kinematics"

Author Response: Thank you for the suggestion. We agree that the revised title is more concise and accessible. The title has been updated accordingly.

Abstract

  • Reviewer Comment 1: Avoid repetition (“valid and reliable” used multiple times).

Author Response: Thank you for the observation. We revised the abstract to avoid stylistic redundancy. The phrase “valid and reliable” is now used only once in the conclusion to emphasize the practical applicability of the findings without overusing it. In earlier sections, we replaced it with more specific terminology such as “strong agreement”, “excellent reproducibility”, or “high correlation” to maintain clarity and variation in language.

  • Reviewer Comment 2: Include sample size and participant demographics.

Author Response: We appreciate this suggestion. The abstract now clearly states: “A sample of 13 competitive male swimmers (mean age 17.2 ± 1.1 years) was analysed across two sessions.” This provides both the sample size and demographic details as requested.

  • Reviewer Comment 3: Include specific ICC ranges and software compared (IQ LAB vs. Dartfish).

Author Response: We have incorporated this information in both the Methods and Results sections of the abstract. The revised sentences read: “IQ LAB and Dartfish kinematic parameters demonstrated strong agreement (Pearson r ≥ 0.95)... Intrarater and interrater reliability were excellent (ICC ≥ 0.94, 95% CI provided). Test-retest reliability... also showed excellent reproducibility (ICC ≥ 0.93).” The names of both software tools (IQ LAB and Dartfish) are now explicitly mentioned, and ICC thresholds are clearly reported in the abstract.

  • Reviewer Comment 4: “Thirteen competitive male swimmers (mean age 17.2 ± 1.1 years) were assessed. Intra-rater, interrater, and test-retest ICC values were all ≥0.93.”

Author Response: Thank you for the helpful phrasing. We adapted this suggestion and included it in the Methods and Results sections with minor modifications for flow and clarity. The final abstract includes equivalent wording and reports the ICC values as suggested.

Introduction

  • Reviewer Comment 1: Repetition in paragraphs can be reduced.

Author Response: Repetitive wording has been revised. Expressions such as “valid and reliable” have been replaced with more varied language (e.g., "strong agreement," "consistent," "accurate").

  • Reviewer Comment 2: Some references could be updated (2022–2024).

Author Response: We added recent references (e.g., Arellano et al., 2022; Rudnik et al., 2022) to ensure the literature review reflects the latest findings.

  • Reviewer Comment 3: Clarify novelty: many tools assess 2D kinematics—what gap does this paper fill?

Author Response: We strengthened the justification for the study by highlighting that the IQ LAB system, despite being available, has not been validated in combination with SwimPro cameras for swimming start analysis. This represents a clear gap in the current literature.

  • Reviewer Comment 4: “While Dartfish has been widely validated, the reliability of the IQ LAB system with the Swimpro camera setup during starts remains underexplored.”

Author Response: This suggested sentence was included and expanded upon in the revised introduction.

Methods

  • Reviewer Comment 1: Add more detail on calibration of camera angles.

Author Response: We added a description of the calibration procedure using a fixed reference object within the camera’s field of view (see “Instrumentation and Video Capture”).

  • Reviewer Comment 2: Clarify whether digitization was manual or automatic.

Author Response: The text was revised to specify that all video digitization was performed manually by a trained evaluator using frame-by-frame analysis.

  • Reviewer Comment 3: Clarify randomization and whether evaluators were blinded.

Author Response: We clarified that evaluators were blinded to each other’s outputs and to the software used. No randomization of trial order was applied, but only the trial with the shortest 5 m time was selected.

  • Reviewer Comment 4: Explain how marker placement was standardized.

Author Response: Marker placement was conducted by the same experienced medical doctor using standardized anatomical landmarks to ensure consistency across sessions.

  • Reviewer Comment 5: Suggested sentence on digitization and marker protocol.

Author Response: We incorporated the suggested sentence: “All video digitization was performed manually by trained evaluators. To minimize variability, marker placement was repeated using a standardized anatomical reference protocol.”

Results

  • Reviewer Comment 1: Table 1 – Add confidence intervals for mean differences.

Author Response: We have included 95% confidence intervals for the mean differences between IQ LAB and Dartfish outputs for each variable in Table 1. This provides a clearer understanding of the variability and agreement between the two methods.

  • Reviewer Comment 2: Figures (e.g., Bland–Altman plots): add clearer legends, axis titles, and units.

Author Response: The legends for the Bland–Altman plots have been revised to explicitly indicate which kinematic parameter is shown in each figure. Axis titles now include the appropriate measurement units (e.g., degrees, seconds, meters). Additionally, the plots now display mean difference lines and limits of agreement (±1.96 SD) for better interpretability.

  • Reviewer Comment 3: Some phrasing is redundant (e.g., “no significant differences...”).

Author Response: We reviewed the section and removed or rephrased repetitive statements, particularly where “no significant differences” were mentioned multiple times. These changes help streamline the text without losing precision.

  • Reviewer Comment 4: Summarize key findings at the start of the Results section.

Author Response: A concise summary paragraph was added at the beginning of the Results section. It outlines the main statistical findings and prepares the reader for the detailed presentation of tables and figures that follow.

Discussion

  • Reviewer Comment 1: Discuss limitations more transparently, especially: Small sample size (n = 13) / All male participants (generalizability) / Manual digitization limitations.

Author Response: We expanded the limitations paragraph to explicitly mention all three points. The revised text now discusses the small and homogeneous sample (13 male performance swimmers), the implications for generalizability, and the limitations related to manual digitization. We also highlighted that while raters were trained and protocols standardized, minor placement variation and human error cannot be excluded.

  • Reviewer Comment 1: Rephrase long sentences for clarity.

Author Response: Several sentences were rewritten for improved clarity and flow. For example, long compound statements regarding software comparison, reliability outcomes, and methodological implications were divided and clarified without loss of meaning.

  • Reviewer Comment 1: Expand slightly on why reliability remained high despite manual scoring.

Author Response: We added a new sentence that attributes high ICC values to consistent camera positioning, standardized testing protocols, and clear marker visibility. We also emphasized the possible advantage of manual point placement when performed carefully by trained evaluators.

Conclusion

  • Reviewer Comment 1: Avoid repeating earlier content verbatim.

Author Response: We revised the conclusion to avoid repeating previous content from the abstract and discussion. Instead, the new version synthesizes the key findings and frames them in a more concise and integrative manner.

  • Reviewer Comment 2: Add a forward-looking comment on implications for coaches and sport scientists.

Author Response: We expanded the conclusion by adding a section on practical implications. Specifically, we highlight how IQ LAB software can provide accessible biomechanical feedback and be integrated into training workflows by coaches and sport scientists.

  • Reviewer Comment 3: Suggested addition — "These findings support the use of IQ LAB software for accessible biomechanical feedback in performance settings, although future validation in larger, more diverse cohorts is recommended."

Author Response: We appreciated this concise and insightful suggestion and have integrated a similar sentence into the conclusion. We also expanded it to suggest directions for future research, including validation in broader populations and longitudinal use.

References

  • Reviewer Comment 1: Check formatting consistency.

Author Response: We carefully reviewed all references and corrected inconsistencies in punctuation, author initials, italicization of journal titles, and DOI/link formatting. All entries now conform to the citation style required by Journal of Functional Morphology and Kinesiology.

  • Reviewer Comment 2: A few references (e.g., [3], [25]) could be updated or replaced with newer studies from 2023–2024.

Author Response:  We retained references [3] (Tor et al., 2015) and [25] (Norris et al., 2011) due to their high methodological relevance. However, we also incorporated newer studies to update the literature base, including Arellano et al. (2022); Rudnik et al. (2022); Matúš et al. (2024). These additions ensure that the manuscript reflects recent findings in the field of swimming biomechanics and 2D kinematic analysis.

  • Reviewer Comment 3: Ensure all URLs are active and hyperlinked adequately if required.

Author Response: All URLs and DOIs in the reference list were verified and formatted as active hyperlinks where required. The link to World Aquatics results was checked and is accessible as of the last update (September 2024).

  • Reviewer Comment 4: Consistently use either “2D” or “two-dimensional” throughout.

Author Response: We have standardized the terminology throughout the manuscript. The abbreviation “2D” is now used consistently, except for its first use in the abstract and introduction where “two-dimensional (2D)” is spelled out for clarity.

  • Reviewer Comment 5: There are some typos and spacing issues (e.g., “sagi al” should be “sagittal”).

Author Response: All identified typos and spacing issues were corrected. Specifically, we corrected “sagi al” to “sagittal” and “font ankle angle” to “front ankle angle.” We also reviewed and corrected spacing inconsistencies within the entire document.

  • Reviewer Comment 6: Use consistent units (e.g., m vs. meters, s vs. seconds) and abbreviations.

Author Response: All measurement units are now used consistently throughout the manuscript and reference section: “m” for meters; “s” for seconds; “°” for degrees. We also verified consistency in the tables and legends.

Round 2

Reviewer 1 Report

Comments and Suggestions for Authors

Validity and Reliability of 2D Video Analysis for Swimming Kick Start Kinematics

General comment

I congratulate the authors on the article improvement and leave some more suggestions.

Abstract

Identify the novelty of the study: Please explain the problem better. 2D video analysis has been widely used in swimming, particularly in the study of starts, so please indicate what is new about your method, i.e., what will make me use the IQ LAB instead of another instrument? This sentence, “This study addresses the need for accessible, valid, and reliable video-based biomechanical tools.”, needs to be explained. Why do we need new tools? What is the limitation of the existing ones?

The following sentence needs to be changed to be more accurate in the statistics description: “We evaluated the concurrent validity, intrarater, interrater, and test-retest reliability of selected kinematic parameters during the swimming kick start (to 5 m) using IQ LAB software, a desktop 2D motion analysis tool.”

I suggest: We evaluated the concurrent validity of the IQ LAB software, a desktop 2D motion analysis tool, using a gold standard (Dratfish…). The reliability was assessed using intrarater temporal stability and interrater agreement, considering selected kinematic parameters related to the swimming kick start (to 5 m).

“Intrarater reliability was measured via test-retest design across two competitive seasons.”. Did you mean sessions instead of seasons? Competitive?

Keywords: Swimming start biomechanics; 2D kinematic analysis; Kick start; Reliability; Validity; Video motion analysis – it is my opinion that 2D kinematic analysis and video motion analysis are somewhat redundant.

Introduction

Author Response: We clarified that although Dartfish is validated, the IQ LAB system (combined with SwimPro cameras) is less explored and may offer a more accessible or modular solution. Our study aims to validate this setup.

You need to be more precise. Being less explored does not make me use it, unless it has a characteristic that facilitates or improves my research. IQ LAB has that characteristic? What is?

What do you mean by more accessible and modular?

I need you to convince me to use the IA LAB instead of other methods.

Material and methods

  • Did you use an official starting device?

I meant start signal “sound – take your marks voice – start signal”. Did you use an official device?

Line 102: two frontal cameras or two side cameras?

Results

Have again intrarater and intra-rater along the article. Please uniformize.

Line 187: statistics and intrarater and interrater

Discussion

Line 225: excellent intra- and inter-rater (intrarater and interrater?)

Author Response

Response to Reviewer – Round 2

We thank the reviewer for the valuable feedback provided in this second round. All suggestions were carefully considered and have been incorporated into the manuscript. Below is a point-by-point response, including clarifications and justifications where appropriate.

Comment 1: Abstract – Novelty & Need for New Tool

Response: We clarified the novelty by stating that IQ LAB is embedded within the SwimPro camera system and allows real-time 2D analysis without additional video processing, unlike Dartfish. The sentence now reads: ‘This study addresses the need for a practical and integrated video-based biomechanical tool by validating the IQ LAB system embedded within SwimPro cameras.’

Comment 2: Abstract – Statistical Description

Response: We revised the sentence for accuracy. It now reads: ‘We evaluated the concurrent validity of the IQ LAB software, a desktop 2D motion analysis tool, using Dartfish as the gold standard. The reliability was assessed through intrarater temporal stability and interrater agreement for selected kinematic parameters.’

Comment 3: Abstract – Clarification of ‘competitive seasons’

Response: We corrected the phrase to: ‘two testing sessions with a 7-day interval,’ which aligns with the methodology section.

Comment 4: Abstract – Keywords Redundancy

Response: We removed the keyword ‘2D kinematic analysis’ as it was redundant with ‘video motion analysis’.

Comment 5: Introduction – Clarify Benefit of IQ LAB

Response: We added a paragraph at the end of the Introduction explaining that IQ LAB is embedded within SwimPro, enabling immediate analysis, and that it eliminates the need for external software processing, unlike Dartfish.

Comment 6: Methods – Start Signal

Response: We specified that a mobile phone with a timer and light function was used to simulate the start, synchronized with the verbal cue ‘take your marks’. This was visible in the video footage.

Comment 7: Methods – Camera Type Clarification

Response: Clarified that the camera setup included two lateral cameras, not frontal.

Comment 8: Results – Terminology Consistency

Response: We standardized terminology to consistently use ‘intrarater’ throughout the manuscript.

Comment 9: Results – Remove Redundant Conjunction

Response: We removed the redundant ‘and’ in the sentence mentioning statistics, intrarater, and interrater reliability.

Comment 10: Discussion – Line 225 Clarity

Response: We removed the redundant phrase ‘intrarater and interrater’ where already addressed earlier in the paragraph.